# Preparation and Characterization of Multilayer pH-Responsive Hydrogel Loaded *Ganoderma lucidum* Peptides

**DOI:** 10.3390/foods12071481

**Published:** 2023-03-31

**Authors:** Ruobing Liu, Jing Gan, Mengdi Du, Xiao Kong, Chunxia Xu, Yue Lü, Shengliang Cao, Ting Meng, Bo Wang, Tianying Yu

**Affiliations:** College of Life Science, Yantai University, Yantai 264000, China

**Keywords:** hydrogel, pH-sensitive, antioxidant efficiency, peptide oral administration, ROS

## Abstract

To develop a safe, targeted, and efficient assembly of a stable polypeptide delivery system, in this work, chitosan, sodium alginate, and sodium tripolyphosphate were used as materials for the preparation of hydrogels. M-SCT hydrogels were prepared by ionic gelation and the layer-by-layer (LBL) method. The composite hydrogels exhibited excellent pH sensitivity and *Ganoderma lucidum* peptides (GLP) loading capacity. The prepared hydrogels were characterized and evaluated. The internal three-dimensional network structure of the hydrogel was observed by scanning electron microscopy (SEM), and Fourier transform infrared (FT-IR) spectroscopy confirmed the electrostatic interactions among the components. X-ray diffraction (XRD) was used to observe the crystal structure of the hydrogel. The maximum peptide encapsulation efficiency was determined to be 81.73%. The digestion stability and thermal stability of M-SCT hydrogels loaded GLP were demonstrated to be improved. The amount of peptides released from the GLP/M-SCT-0.75 hydrogels in simulated gastric fluid was lower than 30%. In addition, the ABTS assays showed that the free radical scavenging ability of the GLP/M-SCT-0.75 hydrogels confirmed the efficacy of the hydrogels in retaining the antioxidant activity of GLP. The study suggested the M-SCT-0.75 hydrogels had a great deal of potential as a peptide carrier for oral delivery.

## 1. Introduction

Medicinal mushrooms are considered to be powerful medicinal fungi that contain many bioactive proteins. Basidiomycetous fungi, such as *Ganoderma lucidum*, have been used for thousands of years in Eastern countries for the treatment and prevention of various diseases. To date, more than 38 fungal immunomodulatory proteins have been identified in *Ganoderma lucidum* [1], giving it a wide range of pharmacological properties, such as antioxidation [2] and anti-aging effects [3], protection against liver cell damage [4], and a possible role in the treatment of cancer [5]. Among them, *Ganoderma lucidum* peptides (GLPs) are some of the foremost practical components isolated from *Ganoderma lucidum*. Compared with the protein and polysaccharide fractions of *Ganoderma lucidum*, GLP has stronger antioxidant function and free radical scavenging activity, protecting the biomolecules from free radical damage [6].

Oxidative stress occurs when there is an imbalance between free radical products and their antioxidant capacity. Excessive amounts of reactive oxygen species (ROS) can damage cells and inhibit their normal function [7]. Oxidative stress is associated with many human diseases as well as the aging process [8], and in severe cases, it can lead to a variety of life-threatening pathological conditions. In such cases, antioxidants need to be applied, which can compensate for increased oxidative stress and regulate physiological and pathological redox. GLP, as a natural antioxidant, has strong antioxidant functions [9] and have attracted a great deal of attention for their use in protecting biomolecules from free radical damage.

However, according to previous reports, peptides are extremely sensitive to gastrointestinal digestion. They are readily hydrolyzed by proteases in the body, which may lose their structural integrity and function [10]. This is also the main challenge faced by commercial products of bioactive peptides. Therefore, oral peptide delivery systems have become an important technology for improving the stability of bioactive peptides.

Given the stable structure, abundant properties, and low price of polysaccharides, they are usually ideal encapsulation materials. Chitosan (CS) is the *N*-deacetylation product of chitin. Its molecular structure contains many hydroxyl, acetylamino, and amino groups, and it is a natural cationic polysaccharide [11]. The protonation of amino groups in an acidic environment, which gives chitosan a positive charge, allows cross-linking with the multivalent anion [12], such as sodium tripolyphosphate (STPP). The advantage of the ionic gelation method using STPP as a cross-linker agent is that it requires only mild conditions that do not damage sensitive proteins and are regarded as suitable systems for protein or peptide entrapment and delivery [13]. CS has potential applications in oral delivery systems due to its good biocompatibility, non-toxicity, pro-permeability, and its ability to enhance drug absorption. Sodium alginate (SA) is a natural linear polysaccharide composed of (1,4)-β-D-mannuronic acid (M) and α-L-guluronic acid (G), which is found in brown algae and marine organisms [14]. SA has good biocompatibility [15] and pH sensitivity [16], and is widely used in the field of biomedicine, especially in sustained-release drug delivery systems.

Previously, researchers have reported that the hydrogels prepared using SA and CS have pH sensitivity and inhibit the release of functional components at low pH values, showing great potential as carriers for drug delivery systems [17,18]. Our interest lies in whether hydrogels prepared using SA and CS could improve the biological stability of antioxidant peptides. To construct a stable pH-responsive delivery system for orally administered *Ganoderma lucidum* peptides, the pH sensitivity of the composite hydrogel needed to be further enhanced, thus preventing the massive release of the loaded GLP in gastric juice and improving the biological stability of the GLP. Therefore, this study aims to prepare a 3D network composite hydrogel using CS, SA, and STPP. By observing the swelling behavior of the hydrogel and the internal network structure, the most pH sensitive sample with a dense internal structure was selected to encapsulate the antioxidant peptide GLP, and the coating of the SA matrix was performed to develop a stable and pH-responsive peptide delivery system. Different analytical tools, such as scanning electron microscopy (SEM), Fourier transform infrared (FT-IR) spectroscopy, and X-ray diffraction (XRD), were used simultaneously to determine the physicochemical properties of hydrogels. Similarly, the in vitro release kinetics and antioxidant properties of free and embedded peptides were studied.

## 2. Materials and Methods

### 2.1. Materials

Chitosan (CS, degree of deacetylation > 85%) was provided by Shandong Haidebei Biotechnology Co., Ltd. (Shandong, China). Sodium alginate (SA, viscosity: 7000–10,000 mPa·s) and sodium tripolyphosphate (STPP) were purchased from Shanghai Macklin Biochemical Co., Ltd. (Shanghai, China). *Ganoderma lucidum* peptides (GLP, 98 wt%) were obtained from Xi’an Lander Biotech Co., Ltd. (Xi’an, China). Pepsin, pig bile powder, and trypsin were supplied by Shanghai Yuanye Bio-Technology Co., Ltd. (Shanghai, China). An ABTS radical scavenging activity assay kit was purchased from Beijing Solarbio Science & Technology Co., Ltd. (Shanghai, China). The simulated gastric fluid (SGF, pH 1.2) was prepared by dissolving 0.6 g NaHCO_3_, 3.1 g NaCl, 1.1 g KCl, and 1 g pepsin in 1 L deionized water. The simulated intestinal fluid (SIF, pH 7.4) was composed by dissolving 2.7 g NaCl, 0.31 g KCl, 10 g trypsin, and 20 g pig bile powder in 1 L deionized water. Calcium chloride and other reagents were obtained from Sinopharm Chemical Reagent Co., Ltd. (Shanghai, China).

### 2.2. Preparation of Hydrogels and the Exploration of Cross-Linker Ratio

#### 2.2.1. Preparation of M-SA/CS/STPP Hydrogels

In this study, the M-SA/CS/STPP (M-SCT) composite hydrogels (Figure 1) were prepared by ionic gelation and the layer-by-layer (LBL) method, according to Feng [19]. The specific methods are as follows: specific amounts of sodium alginate solution were dispersed in distilled water under magnetic stirring for 2 h. Then, a certain amount of CS was added to the SA solution with continuous stirring for 3 h, followed by the addition of STPP powder and further stirring until it was entirely homogeneous. The mass ratios of SA and CS in solution were controlled at 9:1, 8:2, 7:3, 6:4, 5:5, 4:6, 3:7, 2:8, and 9:1. The concentration of STPP was controlled at 0, 0.25, 0.5, 0.75, and 1.0 wt%. Vacuum defoaming of the obtained SA/CS/STPP solution was conducted for 2 h, and then withdrawn via a 5 mL medical syringe. Next, 1 mL of dilute hydrochloric acid was evenly sprayed on the surface of the mixed solution and slowly oozed downward until the solution was converted to a hydrogel [20]. Afterward, the resultant hydrogels were washed with plenty of distilled water to remove residual hydrochloric acid, and the SCTs’ hydrogels were obtained. The hydrogels were cut into 5 mm diameter and 5 mm height hydrogel tablets. Then, during the coating process, selected SCT hydrogel tablets, which had the most pH sensitive and dense internal network structure as core tablets, were immersed in SA (1.5% *w*/*v*) solution for 1 min and placed in a CaCl_2_ solution for 5 min to form a one-layer M-SCT hydrogel. The hydrogel tablets were removed from the mixture and washed in the residual CaCl_2_ with distilled water. The coating process was repeated two times until the 2-layer M-SCT hydrogel could be prepared. For the preparation of the GLP/M-SCT hydrogel, prior to coating, GLP was immobilized on M-SCT hydrogel using the swelling adsorption method. The remaining solutions were prepared using the methods described above.

#### 2.2.2. Determination of the Gelling Strength

The 100 mL beaker was placed on the TMS-Pilot texture analyzer (TMS-PILOT, Food Technology Corporation, Sterling, VA, USA) and P/0.5R was selected as the test probe. The pre-test rate and test rate were 0.5 mm/s, the post-test rate was 1 mm/s, the trigger force was 0.3 N, and the compression distance was 20 mm. The maximum force when the gel was broken was recorded, and the unit was g, which was converted to N/cm^2^ based on the cross-sectional area of the probe.

#### 2.2.3. Evaluation of Encapsulation Efficiency and Loading Capacity

In our study, encapsulation efficiency was used as the crucial parameter for the ratio of STPP in the composite hydrogel. Some modifications were made according to Wang [21]. The GLP/SCTs hydrogel tablets (0.50 g) were submerged for 24 h in a 50 mL centrifuge tube containing 20 mL of pH 7.4 phosphate-buffered solution (PBS). The soaked hydrogel was crushed with a pestle, then 10 mL of fresh PBS was added to the centrifuge tube, sonicated for 30 min, and centrifuged (Centrifuge Pico 17, Thermo Fisher Scientific, Waltham, MA, USA) at 12,000× *g* rpm for 2 min. Next, 3 mL of the supernatant was withdrawn, and the amount of GLP was determined via biuret colorimetry under a UV spectrophotometer (TU-1900, Beijing Purkinje General Instrument Co., Ltd., Beijing, China). The EE and LC of GLP/M-SCT hydrogels were calculated as follows:EE% = (W_Encapsulated GLP in hydrogel_/W_The total mass of GLP_) × 100%(1)
LC% = (WE_ncapsulated GLP in hydrogel_/W_The total mass of hydrogel_) × 100% (2)

### 2.3. Swelling Behavior

To track the swelling behavior of the hydrogel in simulated fluid, the hydrogels were investigated in vitro in different pH environments: SGF, pH 1.2 and SIF, pH 7.4. The freeze-dried hydrogel tablets were dipped in the swelling medium at 37 °C, shaken (100 rpm) for 2 h in the SGF, and shaken for 4 h in the SIF. In this procedure, the swollen hydrogel tablets were removed using a medicine spoon (every 30 min) and weighed after carefully wiping off the liquid on the surface of the hydrogel tablets with filter paper. The swelling ratio was calculated as follows:Swelling ratio (%) = (W_t_ − W_d_)/W_d_ × 100%(3)
where W_t_ is the weight of the swelling hydrogel tablets and W_d_ is the weight of freeze-dried hydrogel tablets.

### 2.4. In Vitro Release Studies

In vitro release experiments were studied using the dried GLP/M-SCT composite hydrogel tablets, which were immersed in 30 mL of different simulated fluids: SGI at pH 1.2 for 2 h and SIF at pH 7.4 for 4 h. Then, the tablets were gently shaken in a thermostated shaker bath at 37 °C and 100 rpm. During the simulated gastrointestinal digestions, a 3 mL solution was collected from the beaker at appropriate intervals (every 30 min in SGF and every 60 min in SIF). The concentration of GLP released at 545 nm was measured by UV-Vis spectrophotometer and 3 mL fresh solution was replenished at the same time.
M_i_ (%) = ∑C_i_ + C_i−1_V_S_(4)
where C_i_ is the concentration of released peptide in the solution at the time i, V is the total volume of release solution, and V_S_ is the sample volume.

### 2.5. Characterization of Hydrogels

The internal morphologies of freeze-dried hydrogels coated with a thin layer of gold were observed using a scanning electron microscope (Gemini SEM 500, Zeiss Ltd., Oberkochen, BW, Germany) at 5 kV. The infrared spectra of control samples and composite hydrogels were recorded by an infrared spectrophotometer (Vertex 70, Bruker Ltd., Ettlingen, BW, Germany). The wavelength range covered 4000 to 500 cm^−1^, with a scan rate of 32 times per second. The crystalline structure of the samples was analyzed using an X-ray diffractometer (XRD) (D8 Advance, Newport Scientific Ltd., Warriewood, SDY, Australia). The X-ray diffractograms of the samples from X-rays at 2θ were recorded in a range of 5° to 80° at a scanning rate of 2°/min. The thermal characteristics of the samples were evaluated using TGA2 (Mettler Toledo, Columbus, OH, USA). Under a 50 mL/min nitrogen flow, samples were heated at a rate of 10 °C/min from 28 °C to 600 °C.

### 2.6. Detection of Free Radicals

Next, 2,2′-azino-bis-3-ethylbenzothiazoline-6-sulfonic acid (ABTS) scavenging activity was evaluated according to Tan [22]. The same volume of a 7 mM ABTS stock solution was mixed with a 2.45 mM potassium persulfate solution to obtain the ABTS solution. The mixture was stored at room temperature in the dark for approximately 12 h before use. It was then diluted with PBS until the solution reached an absorbance of 0.70 ± 0.02 at 734 nm. Next, 8 μL of the mixture of the hydrogel solution was added to 200 μL of the ABTS working solution and incubated in the dark for an additional 10 min at room temperature until the solution reached an absorbance of 734 nm. L-ascorbic acid was used as a positive control. The results were calculated as the percentage inhibition according to the following formula: ABTS radical scavenging activity (%) = A_b_ − (A_s_ − A_c_)/A_b_ × 100%(5)
where A_b_ is the absorbance of the blank, A_c_ is the absorbance of the control, and A_s_ is the absorbance of the sample after reaction.

### 2.7. Statistical Analysis

The SPSS 19.0 program (GraphPad Software, San Diego, CA, USA) and Origin Pro 9.0 (Origin Lab, Northampton, MA, USA) were used to conduct the statistical analyses of the data. All indicators were carried out at least in triplicate, and the data were expressed as the mean value ± SD. The data were considered significant if *p* < 0.05.

## 3. Results

### 3.1. The Exploration of Cross-Linker Ratio

#### 3.1.1. Determination of the Gelling Strength

We researched the effect of the SA:CS mass ratio on the gelling strength of the SCT-0 hydrogel. As shown in Figure 1, with the decrease in the mass ratio of SA to CS, the hydrogel strength increased initially, and then decreased. The strength of the SCT-0 hydrogel peaked when the mass ratio of SA:CS was 6:4. This result is similar to that of the sodium alginate/polyvinyl alcohol polymer [23]. Interestingly, samples with SA:CS mass ratios of 2:8 and 1:9 were not wholly in a hydrogel state due to low cross-linking. Consequently, based on the results in Figure 1, an SA:CS mass ratio of 6:4 was selected to prepare the SCT-0 hydrogel for further studies. 

#### 3.1.2. Evaluation of Encapsulation Efficiency

In this paper, we took the mass concentration of STPP as the variable with which to evaluate the GLP encapsulation efficiency of the SA/CS/STPP (SCT) hydrogel. Table 1 shows that the encapsulation efficiency (%) of the SCT-0 hydrogel was 58.22%. As no significant interactions were observed between GLP and the SA or CS molecular chains, the encapsulation efficiency was primarily determined by GLP adsorption carried out by the pores on the hydrogel. This is similar to the principle of konjac dextran/sodium alginate microspheres [24]. The encapsulation efficiency was increased to 81.73% with the STPP increased from 0 to 1. Compared with the SCT-0 hydrogel, the GLP encapsulation efficiency was enhanced dramatically (*p* < 0.05) with the addition of STPP. 

### 3.2. Analysis of Swelling Behavior

The effect of STPP amounts on the swelling behavior of SCTs’ hydrogels in SGF (pH 1.2) and SIF (pH 7.4) media is presented in Figure 2A. The swelling ratio of hydrogels decreased with the increase in STPP concentrations in SCTs’ hydrogels during the first two hours. In other words, the hydrogel with a higher STPP mass concentration showed a lower degree of swelling in SGF. This phenomenon was in agreement with STPP-anion-cross-linked chitosan microspheres [25]. The trend of swelling in the cross-linked hydrogels is known to be closely related to the drug-release trend. Therefore, adding STPP to hydrogels with the addition of STPP decreased the swelling ratio at a low pH, which would most likely decrease the release of GLP in SGF. We also found that the swelling degree of the hydrogel with increased STPP content was more pH sensitive when the SCTs were moved from the pH 1.2 to pH 7.4 medium. The swelling ratio was in the range of 919.7–2113.3% and 954.7–1959%, respectively, when the STPP concentration was 1 wt% and 0.75 wt%. With the STPP concentration decreasing from 1 wt% to 0 wt%, the swelling ratio of the SCT-0 hydrogel ranged from 1653% to 2020%, so the composite hydrogels prepared with the STPP anion cross-linker were more pH sensitive. 

Figure 2B describes the swelling behavior of SCT-0.75 and M-SCT-0.75 hydrogels. The coating process of M-SCT-0.75 significantly reduced the swelling rate of hydrogels in the acidic environment and improved the pH sensitivity of hydrogels due to the good resistance of multilayer SA matrix’s strong gastric acid resistance [26]. Overall, the M-SCT-0.75 composite hydrogel showed a slower rate of swelling compared to the SCT-0.75 composite hydrogel. In SGF (pH 1.2), the swelling ratio of SCT-0.75 hydrogels was 996.7% after 2 h and the swelling ratio of M-SCT-0.75 hydrogels was 269.3%. In SIF, at the fourth hour, the outer film of M-SCT-0.75 gradually began to dissolve, and the swelling ratio of M-SCT-0.75 was significantly lower than SCT-0.75. Finally, the outer film of M-SCT-0.75 almost disintegrated in the simulated ileum fluid, exposing the hydrogel core, and extending the contact time between the hydrogel core and the small intestine, thereby increasing GLP absorption. In conclusion, the addition of STPP and the coating process had a significant positive effect on the swelling behavior of the composite hydrogel. 

### 3.3. In Vitro Drug Release

To compare the release profiles of GLP/SCT-0.75 and GLP/M-SCT-0.75 hydrogels, GLP release in the simulated gastrointestinal tract was measured. Figure 3 shows the amount of GLP released from the hydrogel at different times. After 2 h of incubation in SGF, 57% and 28% of the GLP were released from the SCT-0.75 hydrogels and M-SCT-0.75 hydrogels, respectively. The amount of GLP tested in the release buffer of the M-SCT-0.75 hydrogel was lower than that released by the SCT-0.75 hydrogel. This is because the trend of swelling in the cross-linked hydrogels is closely related to the trend of drug release [27]. The coating process of M-SCT-0.75 significantly reduced the swelling ratio of hydrogels and improved the pH sensitivity of hydrogels in acidic environments. Correspondingly, the disintegration of the multilayer SA hydrogel resulted in a significant release of GLP in SIF. The total amount of GLP released markedly increased to 77.3%. Compared with the SCT-0.75 hydrogel, the release trend of the M-SCT hydrogel was the same, but the release of the GLP in the M-SCT hydrogel was slower in the simulated gastric fluid. Overall, M-SCT hydrogels effectively reduced the release rate of GLP in SGF, demonstrating that M-SCT hydrogels are potential peptide carriers.

### 3.4. Characteristics of Hydrogels

#### 3.4.1. SEM Characterization and Analysis

The morphology of freeze-dried hydrogels was observed using a scanning electron microscope. As shown in Figure 4, the M-SCT-0.75 possessed a coarser surface with comparatively smaller pores. For SCTs’ hydrogels in Figure 4, the porous structure densified with the increase in STPP in hydrogels, as the introduction of STPP increased the cross-linking point. The porous structure provided more space and could facilitate the entry and storage of GLP, which was in agreement with the results obtained from the encapsulation efficiency of hydrogels. However, when the STPP amount increased to 1.0 wt%, the internal network structure of the SCT-1 hydrogel appeared hollow. This was mainly attributed to the agglomeration of excess STPP [20], which led to increased repulsion of the carboxylic acid anion of SA and the multivalent anion of sodium tripolyphosphate in the whole hydrogel system.

#### 3.4.2. Interaction among SA, CS, and STPP

The chemical structure between SCTs and M-SCT is shown in the FTIR spectra (Figure 5). In the spectra, the absorption bands of CS at 1656 cm^−1^ and 1605 cm^−1^ represented amide I the (C=O stretching of the secondary amide) band and amide II bands (the N-H stretching of the secondary amide). There was a vibration peak of C-N at 1156 cm^−1^, and 3429 cm^−1^ was the absorption band of N-H and O-H stretching vibration, reflecting the hydrogen bond between the hydroxyl and amino groups in the chitosan molecule. The protonated carboxyl groups (-COOH) in SA were observed as characteristic peaks at 1614 cm^−1^. The characteristic peaks observed at 1212 cm^−1^ were the P=O in STPP [28]. Furthermore, the bands at 896 cm^−1^ and 736 cm^−1^ belonged to the asymmetric and symmetric stretching modes of P-O-P linkages, respectively [29].

However, the strength of the peak at 1156 cm^−1^ was weakened in the spectrum of SCT-0 hydrogels, indicating that the ionic interaction between SA and CS occurred in the positively charged amino groups (-NH_3_^+^) of CS [30]. Meanwhile, the amino peak at 1156 cm^−1^ disappeared in the spectrum of SCT-0.75 hydrogels, and the disappearance of the band demonstrated the ionic cross-linking of chitosan [31] and phosphoric acid, proving that the introducing STPP increased the cross-linking points. In SCTs and M-SCT composite hydrogels, the band shifted to the low frequency at 3354 cm^−1^ and 3414 cm^−1^, respectively, indicating a strong electrostatic interaction between CS, SA, and STPP molecular chains. An interaction of CS with STPP was also observed between 896 and 736 cm^−1^; this range is characteristic of the symmetric stretching pattern of these bonds with respect to (p-o-p) and o-p-o deformations. Based on the above analysis, it was proven that the SCTs and M-SCT hydrogels were formed.

#### 3.4.3. XRD Analysis of Hydrogels

The results of the research on the crystal structure of hydrogels by XRD are shown in Figure 6. The successful cross-linking of CS, SA, and STPP was confirmed by XRD. The diffraction peak at 2θ = 14.3° to 21.7° of the SA scaffold showed the typical amorphous structure of SA, and the XRD pattern of native chitosan revealed two peaks at 2θ = 11.5° and 19.9°. The results were similar to that of alginate/gelatin in reference [32]. 

The diffractogram of the hydrogel showed that the diffraction peak of CS disappeared at 11.5°; this can be explained by the strong interaction between CS and SA as well as STPP, which destroyed the crystalline structure. After cross-linking with SA and STPP to CS, the diffraction peak at 20.4° broadened for SCT-0 and at 20.8° for SCT-0.75. M-SCT-0.75 diffraction peaks were broad and diffuse with reduced crystallinity, reflecting the restricted motion of CS chains in intramolecular interactions. Meanwhile, the introduction of SA and STPP hindered the formation of intramolecular hydrogen bonds associated with the CS [33]. 

In summary, the crystal structures of these three materials were broken, and more amorphous morphology was formed [34]. This phenomenon also indicated the existence of good compatibility between the composite hydrogel components due to the strong interaction between CS, SA, and STPP.

#### 3.4.4. Thermostability of the Hydrogels

Research on the thermal stability of pure materials and hydrogels is presented in Figure 7. The TGA curve of pure SA showed that the initial thermal weight loss below 100 °C was mainly caused by the loss of water evaporation [35]. The rapid thermal weight loss of pure SA occurred around 200 °C, which was caused by the breakage of mannuronic and glucuronic acid fragments [36,37]. Similarly, the slow thermal weight loss of CS around 100 °C was attributed to the loss of moisture, while its major degradation occurred above 230 °C, indicating that CS showed better thermal stability than SA. Similarly, the significant degradation temperature for pure GLP was approximately 150 °C, demonstrating worse thermal stability than that of pure SA or CS. 

Figure 7A shows that the thermal decomposition process of SCT-0.75 hydrogels included three stages in the temperature range of 28 °C–600 °C. The initial mass loss of the SCT-0.75 hydrogel occurred between 28 °C and 100 °C, and a large amount of water was lost in this temperature range. The degradation temperature for hydrogels was approximately 150 °C, which was lower than that of the pure ingredients. The implication was that the thermal stability pattern of the SCT-0.75 composite hydrogel was worse compared to that of pure SA or CS. Previous studies on the sodium alginate/carboxymethyl chitosan hydrogels produced similar results [38]. The significant weight loss in the second stage in the temperature range of 150 °C–280 °C was attributed to the disintegration of intra and intermolecular interactions, the depolymerization of molecular rings, and the cleavage of saccharide rings and glycosidic bonds [39]. In the last stage, the hydrogel underwent thermal decomposition in the temperature range of 300 °C–500 °C [38]. 

As shown in Figure 7B, the DTG curves exhibited that the decomposition of hydrogels’ rates decreased due to the addition of Ca^2+^ or GLP. Compared with SCT-0.75 hydrogels, M-SCT-0.75 hydrogels and GLP/M-SCT-0.75 hydrogels exhibited better stability. These results suggest that the thermal stability of M-SCT-0.75 hydrogels is better than that of SCT-0.75 hydrogels because the cross-linking of Ca^2+^ increases the force between SA macromolecules [40,41]. Furthermore, the values of the total weight loss of SCT-0.75 hydrogels, M-SCT-0.75 hydrogels, and GLP/M-SCT-0.75 hydrogels during the three stages were 74%, 55%, and 50%, respectively. M-SCT-0.75 hydrogels and GLP/M-SCT-0.75 hydrogels exhibited greater residual quality. Complex polymers of polysaccharides and peptides could improve the stability of peptides during thermal denaturation.

### 3.5. Antioxidant Activity of the Cross-Linked Hydrogel

The ABTS free radicals have been widely used to evaluate antioxidant materials’ ability to scavenge free radicals. ABTS can be oxidized by K_2_S_2_O_8_ to ABTS^+^ free radicals, which show a green color in solution and exhibit significant absorption at 734 nm. Antioxidants can burst the ABTS^+^ radical and convert it to a lighter color, resulting in a decrease in absorbance at 734 nm. GLP is one of the natural antioxidants; therefore, the free radical scavenging ability of the GLP and GLP/M-SCT-0.75 hydrogels was first explored using ABTS assays. As depicted in Figure 8, the radical scavenging activity of GLP encapsulated in the M-SCT-0.75 hydrogel was compared with that of pure GLP, and the effect of the encapsulation process on the antioxidant capacity of GLP was evaluated. The activity of GLP in scavenging ABTS radicals increased with increasing peptide concentration (1–8 mg/mL) in the range of 19.63–69.83%, which is similar to the result of the kilka-derived peptidic fraction [28]. In addition, the scavenging activities of the GLP and GLP/M-SCT-0.75 hydrogels did not differ significantly, proving that the encapsulated peptide still has a strong antioxidant capacity. 

## 4. Discussion

The natural antioxidant GLP was previously reported to have a strong antioxidant capacity; however, the low biological stability of GLP limits its practical application. Therefore, in this experiment, we constructed the M-SCT composite hydrogels using the internal gelation and layer-by-layer (LBL) methods. Through structural characterization and antioxidant identification experiments, it was confirmed that the M-SCT composite hydrogels had a stable embedding ability of GLP and enhanced peptic tolerance at the same time.

In this study, the cationic amino group of CS interacted electrostatically with the carboxylic acid anion of SA and the phosphate ion of STPP in an acidic environment to form a composite SCTs hydrogel. We began by optimizing the ratio of SA to CS. The optimal ratio of SA and CS was selected with the goal of gel strength, and the results suggested that the best properties were obtained when the SA:CS mass ratio in the hydrogel was 6:4. This may be attributed to the cross-linking density of SA and CS [42]. The carboxylate anions of SA could have electrostatic interactions with the cationic amino groups of CS, leading to the formation of many connections between polymers. The strength of hydrogels also changed according to the degree of cross-linking. However, the SA:CS mass ratio with the optimal features could be different in polymers due to the different degrees of deacetylation, viscosity, and preparation of the polymer.

Meanwhile, the GLP was loaded into the composite hydrogels according to the swelling-adsorption method, which was simple to operate and less likely to cause a loss of GLP in the GLP/M-SCT hydrogel system. Then, the amount of STPP was determined according to the encapsulation efficiency. The results showed that when the amount of STPP was 1, the GLP encapsulation efficiency of hydrogel was the highest, reaching 81.73%. The hydrogels synthesized in this study had a better encapsulation ability compared to konjac glucomannan/sodium alginate/graphene oxide microspheres loaded with ciprofloxacin (EE of 19.11%) [24]. The GLP encapsulation efficiency of hydrogel increased with the gradual addition of STPP. This may be because STPP, as a cross-linker, provides more binding sites that are readily available for interaction, increasing the cross-linking density of the SA/CS hydrogel [25], and promoting the formation of network structure space inside the hydrogel, which is conducive to the storage of GLP. Interestingly, when the mass concentration of STPP was more significant than or equal to one, there were different degrees of folds on the surface of the hydrogel and visualized hollow phenomena. These could be mainly attributed to the polyanions excess of STPP, which had enhanced repulsion with the carboxylate anion in SA. In a previous report on polyphosphateanion/chitosan microspheres, it was also found that the repulsive forces between the anions led to a poor microsphere structure [25]. In addition, in simulated gastric juice, due to the low degree of ionization of the STPP anionic cross-linker at a low pH, hydrogels with more STPP content were more stable and showed a slow rate of swelling in SGF. The results indicated that the addition of STPP was beneficial to the physical properties of the hydrogels. 

In order to reduce GLP loss on the GLP/SCTs’ hydrogel surface while increasing the pH sensitivity of the hydrogels, SA/Ca^2+^ films were wrapped on the hydrogel according to the layer-by-layer coating method. The results showed that after two hours of immersion in simulated gastric juice, the swelling rate of M-SCT-0.75 hydrogels was significantly reduced compared to that of SCT-0.75 by 727.4%. This is due to the pH sensitivity of multilayer SA matrices in acidic environments, demonstrating low swelling properties, while the M-SCT-0.75 hydrogel swelled significantly in an alkaline environment [26]. By simulating the change of pH in the human gastrointestinal tract, we observed that M-SCT could slow down the release of GLP in simulated gastric juice and release a large amount in the intestine, minimizing the loss of GLP. This phenomenon showed that the cross-linked network of M-SCT hydrogels formed by the LBL coating method was more diverse than that of SCTs’ hydrogels, which had a significant positive effect on the swelling properties and the slow-release ability of M-SCT composite hydrogels.

FTIR and XRD were used to examine the cross-linking among CS, SA, and STPP. FTIR results showed that the cationic amino group of CS could interact with the anionic carboxyl group of SA and the phosphate ion of STPP, resulting in the formation of composite hydrogels. Additionally, the conformational analysis by XRD showed good compatibility between the composite hydrogel components. Similarly, we analyzed the stability of GLP/M-SCT-0.75 using TG-DSC. The values of the total weight loss of GLP/M-SCT-0.75 hydrogels during the three stages were 50%, and GLP/M-SCT-0.75 hydrogels exhibited greater residual quality. Therefore, it can be concluded that the thermal stability of hydrogels was improved by the LBL method.

The ABTS tests were performed to verify the in vitro antioxidant activities of the GLP/M-SCT-0.75 hydrogels. We found that the GLP/M-SCT-0.75 hydrogel could scavenge ABTS^+^ radicals in an ABTS radical scavenging ability experiment, and there was no significant difference compared with free peptide, which had a strong antioxidant ability. Therefore, the experiments showed that GLP/M-SCT-0.75 hydrogels prepared according to the internal gelation method and layer-by-layer (LBL) method could significantly improve the biological stability of GLP, and the original antioxidant activity of GLP would remain unaffected.

## 5. Conclusions

In this work, GLP/M-SCT-0.75 hydrogels were constructed via ionic gelation and the LBL method for peptide delivery. The FTIR analysis showed that CS, SA, and STPP were cross-linked by electrostatic interaction. The M-SCT-0.75 hydrogels possessed beneficial properties such as GLP encapsulation, pH sensitivity, and delayed peptide release, and they reduced the contact time of GLP with proteases. The multilayer SA matrix weakened the swelling properties of the hydrogel and improved the pH sensitivity of the hydrogel. Lastly, the ABTS showed that the pure GLP retained its antioxidant activity after incorporation into the M-SCT-0.75 hydrogels. The encapsulation process had no significant effects on the antioxidant capacity of the GLP. Overall, this study demonstrated that M-SCT-0.75 hydrogels may have broad prospects in the delivery of such nutrition components. A more detailed study on the developed peptide delivery systems and their role in delaying and preventing the onset of the oxidation process is in progress.

## Data Availability

This paper presented all relevant data and methods. Any further questions should be directed to the corresponding author.

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
