# Peer review of "Preparation and Characterization of Multilayer pH-Responsive Hydrogel Loaded Ganoderma lucidum Peptides"

_foods, 2023, doi:10.3390/foods12071481_

Round 1

Reviewer 1 Report

Dear Authors

The article entitled “Preparation of Multilayer pH-Responsive Hydrogel Loaded with Ganoderma Lucidum Peptides and its Antioxidant Mechanism” is a very interesting work. Fallowing are few suggestions which might be helpful for the improvement of the article a bit

1. Authors should mention the full forms for the first time for any abbreviation example TBHQ, BHA, BHT.

2. Authors mention about the natural antioxidants beneficial over chemically synthesized antioxidants. Authors explain the side effects or the limitation and the advantages of the natural antioxidants to justify the claim in introduction section.

3.  The introduction section is well written. As the authors mentioned that the hydrogel of the CS-SA and TPP is well established, then authors should explain the novelty of the research in the last paragraph of the introduction.

4. What is M-SA in the section 2.2.1?

5.  For the preparation of the M-SCT, Authors dissolved the CS in the distilled water. As the research the CS is very poorly soluble in water. Authors should check the solubility process of the CS in the solvent system.

 6. Why authors selected CaCl2 as the second crosslinking material rather than the TPP?

7. Authors use the pH 6.8 for the entrapment efficiency study where for the swelling study used the pH 7.4.  Please explain the reason of the selection of the pH with proper references.

8.  Authors claimed that without any cross-linker (SCT-0) the encapsulation 58.22%. Authors must explain the mechanism of the results of the table 1 with proper references 

Reviewer 2 Report

A manuscript presents preparation of an optimal composite hydrogel by chitosan, sodium alginate, and sodium tripolyphosphate for encapsulation and delivery of G. lucidum bioactive peptide Different analytical methods and techniques were used to determine the physicochemical properties of hydrogel. Additionally, in vitro release kinetics and antioxidant properties of free and embedded peptides were studied.

Improving the bioavailability of peptides is essential to improving their bioefficacy.

I suggest some corrections on manuscript:

1. Title, line 3: …….its Antioxidant Mechanism. This research does not investigate antioxidant mechanism of the G. lucidum peptides. Only one in vitro screening method for the scavenging ability on reactive species was applied.

To define an antioxidant mechanism of the compound is necessary to apply different antioxidant methods with different mechanism of action. Antioxidants can demonstrate their properties at different stages of the oxidation process and by different mechanisms: 1) as primary and secondary antioxidants by hydrogen atom transfer, proton coupled electron transfer, radical adduct formation, sequential electron proton transfer, sequential proton loss electron transfer, and sequential proton loss hydrogen atom transfer mechanism; 2) with the potential to interact with various redox signaling pathways by modulation of the activity of redox enzymes and the generation of bioactive secondary metabolites; and 3) with the potential of immune regulatory activity because the disruption of oxidative balance may be linked to the immune system.

e.g. Kozarski, М., van Griensven, L.J.L.D. Oxidative stress prevention by edible mushrooms and their role in cellular longevity, in: S.B. Dhull, A. Bains, P. Chawla, P.K. Sadh (Eds.), Wild Mushrooms Characteristics, Nutrition, and Processing (1st Edition), Taylor & Francis Ltd, London, 2022, pp. 319-348. ISBN: 9780367692513

https://www.taylorfrancis.com/chapters/edit/10.1201/9781003152583-13/oxidative-stress-prevention-edible-mushrooms-role-cellular-longevity-maja-kozarski-leo-van-griensven?context=ubx

2. Lines 30-43: This part of the introduction has already been presented in numerous papers; it is well known that ROS in the body leads to oxidative stress….The focus is on G. lucidum peptides and maybe it will be better to primarily present current understanding of the structure and function of mushroom bioactive proteins and then importance of the antioxidant defence.

e.g.  Liu Y, Bastiaan-Net S and Wichers HJ (2020) Current Understanding of the Structure and Function of Fungal Immunomodulatory Proteins. Front. Nutr. 7:132. doi:10.3389/fnut.2020.00132

3. Lines 45-45: Statement is not clear. Primary role of proteinogenic amino acids in the human body is not to scavenge reactive species. These amino acids are used for de novo synthesis of proteins with different roles as antibodies, enzymes, messengers, hormones, neurotransmitters, transport/storage and structural capacities…. 

4. Lines 52-54: …… and protect the human body from free radical damage. Is the mentioned investigation (reference 8) related to the clinical study?

5. Line 77: Scientists or researchers- should be instead of people.

6. Line 164: Which method was used to measure peptide concentration? It should be mentioned in the manuscript.

7. Lines 380-382 and Fig. 8: Whether the activity of pure hydrogel was measured? May the components of hydrogel have the antioxidant potential? e.g. chitosan

8. Fig. 8: Y axes: It is scavenging potential not rate.

9. Lines: 383-384: Is the antioxidant activity of the encapsulated peptide a result of the peptide itself or additionally of the coating material?

10. Conclusion: It will be good to emphasize that detail conformation analysis of G. lucidum peptide is necessary as well as investigation of its antioxidant mechanisms of action and influence on cells signaling pathways…

Reviewer 3 Report

Overall the paper appears to be a jumble of tests with no coherence and structure toward a set hypothesis. Especially the XRD and TGA tests seem almost arbitrary since they provide minimal information during expected in-vivo situations. No clear comparison to previous studies is made. It is unclear  The stated aim of the authors “Whether the hydro-gels prepared by SA and CS could solve the biological stability and bioavailability of antioxidant peptides caught our attention. Has not really been addressed. Captions are generally insufficient. “ we selected the SCT hydrogel tablets with the optimal features”  What were these? How were the samples selected?

Other issues are:

·        Latin names should be italicized. Why were no enzymes used in in-vitro digestion?

·        Line 77 “people”  - too casual.

·        Line 98 KCl instead of KCL

·        ml or mL?

·        The concentration of what is in the X-axis of Figure 8

Minor mistakes include chemical formulas not shown with subscripts. pH is used with an “=” etc. 

Reviewer 4 Report

The paper was intended to obtain a hydrogel from chitosan, sodium alginate and sodium tripolyphosphate with the purpose of using it as carrier for Ganoderma Lucidum antioxidant peptides.

Although the idea is interesting, there are several aspects that should be addressed in order to ameliorate the article.

 - The aims of the study are clearly expressed.

- The obtained results are concise, rather clearly presented and discussed. References to other published data (where available) are made and satisfactory explanations for the reported outcomes are offered.

  Conclusions are drawn according to the obtained data.

- There are some abbreviations that are not included in the list existing at the end of the paper. Even if some of them (TBHQ …) are easily recognizable, they should be added.

- More details about the reason of selecting Ganoderma Lucidum as source of peptides should be added.

- The experimental program should be described in such manner that it can be easily applied.

- All the reagents used for experiments must be included in the dedicated section along with their providers. Suppliers of ABTS, ascorbic acid, potassium persulfate etc. are missing.

- Information about the name and providers of used apparatus (such as texture meter, centrifuge etc.) should be added.

- Some of the working methods require a careful revision (“The simulated gastric fluid (SGF pH=1.2) was composed of dissolving 2.0 NaCl, and 3.2 g pepsin in 1L deionized water” etc.).

- It is recommended to include a justification for the choice of the encapsulation method.

- A comparison between the prepared hydrogel carrier and other similar materials is necessary in order to sustain the adequacy of the obtained product.

There are multiple phrases difficult (“Antioxidants, as a scavenger, which can reduce the human physiological environment of the process of oxidation to compensate for the increased oxidative stress and regulate the redox of physiology and pathology.” “At present, the antioxidants on the market are mainly through chemical synthesis, and those are effective in reducing oxidation, but pose undesirable side effects, such as TBHQ, BHA, BHT.” “Its molecular structure contains many hydroxyl groups, acetylamino groups, amino groups and other groups, which is a natural cationic polysaccharide.” and so on) to understand or needing rewriting in order to avoid repeated words and formulations (“Therefore, accurate and efficient peptide de livery systems have become an important technology to improve the utilization of food bioactive peptides and promote human health.”…).

Round 2

Reviewer 1 Report

Dear Authors , 

The explanations are well-mentioned. Go through the grammar correction one more time before the final submission. 

Reviewer 2 Report

I have no additional comments

Reviewer 3 Report

during the coating process, we selected the SCT hydrogel tablets with the optimal features as core tablets the optimal features are not described.

Justification for the tests should be given in the introduction.

The captions are still insufficient. Please follow advice from https://www.internationalscienceediting.com/how-to-write-a-figure-caption/

Reviewer 4 Report

The observations were considered and the manuscript adjusted in accordance with the suggestions. 

I would like to congratulate the authors for the work realized for improving the manuscript quality.
